# Bile Acid-Related Regulation of Mucosal Inflammation and Intestinal Motility: From Pathogenesis to Therapeutic Application in IBD and Microscopic Colitis

**DOI:** 10.3390/nu14132664

**Published:** 2022-06-27

**Authors:** Federica Di Vincenzo, Pierluigi Puca, Loris Riccardo Lopetuso, Valentina Petito, Letizia Masi, Bianca Bartocci, Marco Murgiano, Margherita De Felice, Lorenzo Petronio, Antonio Gasbarrini, Franco Scaldaferri

**Affiliations:** 1IBD Unit—UOS Malattie Infiammatorie Croniche Intestinali, CEMAD, Digestive Diseases Center, Fondazione Policlinico Universitario “A. Gemelli” IRCCS, Università Cattolica del Sacro Cuore, L. Go A. Gemelli 8, 00168 Rome, Italy; pgpuca@gmail.com (P.P.); lopetusoloris@libero.it (L.R.L.); valentina.petito@policlinicogemelli.it (V.P.); letizia.masi94@gmail.com (L.M.); antonio.gasbarrini@policlinicogemelli.it (A.G.); franco.scaldaferri@policlinicogemelli.it (F.S.); 2Dipartimento di Medicina e Chirurgia Traslazionale, Università Cattolica del Sacro Cuore, L. Go F. Vito 1, 00168 Rome, Italy; biancab97@hotmail.it (B.B.); murgianeddu@gmail.com (M.M.); margherita.def93@gmail.com (M.D.F.); lorenzo.petr@libero.it (L.P.)

**Keywords:** bile acids, gut microbiome, inflammatory bowel diseases, microscopic colitis, bile acid-induced diarrhea

## Abstract

Inflammatory bowel diseases (IBD) and microscopic colitis are chronic immune-mediated inflammatory disorders that affect the gastroenterological tract and arise from a complex interaction between the host’s genetic risk factors, environmental factors, and gut microbiota dysbiosis. The precise mechanistic pathways interlinking the intestinal mucosa homeostasis, the immunological tolerance, and the gut microbiota are still crucial topics for research. We decided to deeply analyze the role of bile acids in these complex interactions and their metabolism in the modulation of gut microbiota, and thus intestinal mucosa inflammation. Recent metabolomics studies revealed a significant defect in bile acid metabolism in IBD patients, with an increase in primary bile acids and a reduction in secondary bile acids. In this review, we explore the evidence linking bile acid metabolites with the immunological pathways involved in IBD pathogenesis, including apoptosis and inflammasome activation. Furthermore, we summarize the principal etiopathogenetic mechanisms of different types of bile acid-induced diarrhea (BAD) and its main novel diagnostic approaches. Finally, we discuss the role of bile acid in current and possible future state-of-the-art therapeutic strategies for both IBD and BAD.

## 1. Introduction: Basis of Bile Acids Physiology

Bile composition is not limited only to bile acids (BA). In fact, it is composed of water, electrolytes, bile acids, phospholipids, and cholesterol. It is secreted by the liver and plays a role in several functions, including digestion and fat absorption, cholesterol homeostasis, and the maintenance of antimicrobial gut defense [1].

An imbalance in bile production, reabsorption, or functioning is associated with various gastrointestinal affections [2].

### 1.1. Bile Production and Composition

Bile composition is not limited only to bile acids. In fact, if biliary acids represent the majority of bile composition (65–70% approximately), other significant solutes are phospholipids (20–25%), cholesterol (4%), bilirubin, proteins (4%), and other electrolytes and minerals, such as sodium, potassium, chloride, bicarbonate, calcium, phosphate, magnesium, iron, and copper [3].

The synthesis of bile acids begins inside the hepatocytes. At this level, two acids, called “primary bile acids” (PBA) are produced: cholic acid (CA) and chenodeoxycholic acid (CDCA). These two acids are produced from cholesterol following two different pathways:(1).The “classic pathway”, mediated by the Cyp7A1 and Cyp8B enzymes, leads to the production of cholic acid; in this pathway, Cyp7A1 is the limiting step.(2).The “alternative pathway”, mediated by Cyp27 and Cyp7B1, leads to the production of chenodeoxycholic acid [4].

The classical pathway is considered dominant and more represented in adult humans in comparison to the alternative pathway, which is predominant in newborns and children [5].

Before the secretion of bile acids into the bile canaliculi, glycine or taurine is added to the lateral chain of CA or CDCA in a process called conjugation. This is aimed at making primary bile acids more hydrophilic so that they can stay inside the GI tract without being passively absorbed through enterocytes. In this way, they can execute their action inside the gastrointestinal lumen and specific transporters are required for their intestinal reabsorption [6].

Inside the gastrointestinal mucosal lumen, CA and CDCA undergo further processes of modification led by bacterial species of gut microbiota. This process includes several molecular modifications, the most important of which is represented by the 7α/β dihydroxylation. Thanks to this modification, deoxycholic acid (DCA) and lithocholic (LCA) acid are generated, respectively, from cholic acid (CA) and chenodeoxycholic acid (CDCA). Since they derive from the modification of primary bile acids, DCA and LCA are also called “secondary bile acids” (SBA) [7,8]. 7α dihydroxylation is a multistep pathway carried out by different BA-inducible (bai) genes; it seems to belong to different strains, including *Clostridium scindes*, *C. hiranonis*, and *C. hylemonae* [9]. Recently, also *Faecalicatena contorta* S122 has been implicated in the conversion of CA/CDCA to DCA/LCA since it harbors putative BA-inducible operons. Several studies showed that *Eggerthella lenta* is capable of oxidizing and epimerizing BA hydroxyl groups and that it contains “bai-like genes”; therefore, *E. lenta* could also be involved in BA metabolism. With regard to the 7β dihydroxylation, the intervention of *Ruminococcus gnavus* seems to be important [10].

Another significant role is played by the hydrolysis of conjugated bile acids, by which bile salt hydrolase (BSH) splits conjugated bile acids into free bile acids and glycine or taurine through the hydrolysis of the C-24-N-acyl bond that connects bile acids to the conjugated amino acid [8]. This reaction occurs mostly in the lower small intestine and proximal colon where several types of Gram-positive bacteria express BSH including *Clostridium*, *Enterococcus*, *Bifidobacterium*, and *Lactobacillus*; however, for Gram-negative bacteria, the expression of BSH has been found only in members of the genus *Bacteroidetes* [11].

Finally, desulfation must also be mentioned. However, not much is known about the role played by intestinal bacteria in BA desulfation; the bacterial genera responsible include *Clostridium*, *Peptococcus*, and *Fusobacterium*. This reaction mechanism has not yet been elucidated and the enzymes have not yet been characterized [12].

A summary of the biotransformation of BA is given in Figure 1.

### 1.2. The Enterohepatic Circulation

Once secreted by the liver, the bile is accumulated inside the gallbladder; hence, after a meal and under the effect of specific hormones such as cholecystokinin, it is secreted into the gastrointestinal tract. Cholecystokinin, in fact, favors the relaxation of Oddi’s sphincter as well, thus facilitating the delivery of BA and other bile components into the duodenum [13].

Once inside the gastrointestinal tract, BA are widely reabsorbed into the portal circulation and then again secreted into the biliary tract. In part, BA are reabsorbed in part by cholangiocytes and by renal proximal tubule cells. This entire system is very efficient so only 10% of BA delivered into the gastrointestinal tube escape this process. Several transporters and transcription factors are involved in the maintenance of this complex homeostasis. In particular, the transporter ASBT (apical sodium bile transporter), encoded by the gene SLC10A2, is the main channel involved at the GI level, reabsorbing BA coupled with sodium. On the other side of the enterocyte, an Organic Solute Transporter (OST) secretes BA into the portal circulation [14].

Once they have reached the hepatocytes, BA are first transferred, again coupled with sodium, inside the hepatocyte cytoplasm thanks to the transporter NTCP (Na taurocholate cotransporting polypeptide) encoded by the gene SLC10A1; then, they are again actively secreted into the biliary system thanks to the hydrolyzation of an ATP molecule by OATP1B1/B3 [15].

## 2. The Role of BA and Bile Acid Receptors (BARs) in the Pathogenesis of IBD and Dysbiosis

### 2.1. Bile Acids Metabolism

The absorption of the bile acid pool that reaches the gut is usually about 95% efficient. The remaining 5% of BA (0.3–0.6 g of BAs/day) elude absorption and may be metabolized by the intestinal microbiota. As described above, among the biotransformations that BA can undergo are deconjugation (enzymatic hydrolysis of the N-acyl amide bond linking bile acids to their amino acid conjugates) and dihydroxylation (replacement of hydroxyl group with a hydrogen). Deconjugation seems to reduce the bactericidal activity of BA; this reaction, which is substrate-limiting, can take place throughout the entire gastrointestinal tract and is completed in the colon [16].

Many studies have shown that in germ-free mice models there is a reduced diversity of BA species due to the lack of unconjugated and secondary bile acids and an increase in the contents of conjugated bile acid and 3-OH-sulfate bile acids [17,18].

A similar scenario is also detectable in IBD patients because of the dysbiosis present in these subjects. In fact, although in healthy individuals at least 100–150 different intestinal species can be found, in a substantial portion of IBD patients there is a reduction in bacterial diversity associated with an increase in fungi and bacteriophage species [19]. The typical IBD dysbiosis is characterized by an expansion of *Proteobacteria* and *Fusobacteria* and with a reduction in other phyla such as *Firmicutes*, and especially *Clostridiales*, *F. prausnitzii,* and *E. rectalis*. This dysbiosis leads to reduced production of protective and anti-inflammatory metabolites such as short-chain fatty acids, tryptophan metabolites, and Bas [18].

Along with an alteration of gut microbiota, a significant alteration of the BA pool and BA metabolism is detectable in patients with IBD.

Several studies have investigated the composition of the BA pool in IBD patients over the years. In general, these studies demonstrated that especially in Crohn’s Disease (CD) patients, there is a bile acid malabsorption due to ileal involvement, which leads to a reduction in the bile acid pool size when compared to normal subjects. Pioneering studies by Vantrappen et al. demonstrated that the decrease in the bile acid pool size inversely correlates with the Crohn’s Disease Activity Index (CDAI) [20], and Rutgeerts et al. showed that the kinetics of primary bile acids revealed an increased turnover in IBD patients with ileal dysfunction, and the CA loss correlated with the extension of ileal involvement [21]. Several studies have demonstrated that the percentage of conjugated bile acids is increased and the unconjugated and secondary bile acids are reduced in both active CD and UC patients with colonic involvement in comparison with healthy controls. Together, these studies have established that a bile acid malabsorption occurs mainly in CD patients and that the colon has an important role in the biotransformation of primary bile acids into secondary ones [22,23]. Moreover, although the fecal bile acid content is the same in inactive IBD and healthy subjects, during disease the proportions of conjugated BA and 3-OH-sulphate bile acid increase; in addition, the number of secondary BA drops [24]. Similar results have been found by Franzosa et al. in a recent publication. The authors, through an untargeted LC–MS metabolomic and shotgun metagenomic profiling of stool samples of CD, UC patients, and healthy controls, showed a severe reduction in fecal DCA and LCA and an increased abundance of PBA such as CA and CDCA in active IBD patients [18].

A multi-omics profiling of 103 IBD patients showed a significant reduction in SBA, LCA, and DCA and an enrichment of CA and its glycine and taurine conjugates in stool samples of CD patients with dysbiosis compared with those without dysbiosis [25].

Jacobs et al. reported similar data in pediatric IBD patients since they observed an increase in primary bile acids and their conjugated forms, such as CA, CDCA sulfate, and 7-suplohoCA, and also in 7-ketodeoxycholic acid and 3-sulfodeoxycholic in those patients. Even Wang et al. found a significant reduction in secondary BA and an unconjugated BA in 29 pediatric CD patients compared with 20 healthy subjects. These variations in the BA pool correlated with a reduced abundance of bacterial species with BSH and 7alpha-dehydroxylation enzymes, such as *Bacteroides*, *Bifidobacteria*, *Clostridium*, *Eubacterium*, and *Lactobacillus* [26].

Overall, these findings together with data from germ-free mouse models, highlight the paramount role of microbiota in the deconjugation, dihydroxylation, and desulfation of bile acids. Since secondary bile acids act as ligands for TGR5 and other bile receptors with anti-inflammatory and protective properties (discussed later), this reduction could then aggravate the immune dysfunction seen in IBD patients.

### 2.2. Bile Acids Receptors

Among the BAR, increased attention has been focused on farnesoid X receptor (FXR), a nuclear transcription factor activated by primary bile acids, G-protein bile acid-activated receptor (GPBAR)-1 or TGR5, a seven-transmembrane G-protein-coupled receptor, pregnane-x-receptor (PXR), vitamin-D receptor (VDR), and the sphingosine 1-phosphate receptor (SP1R)2, all activated by both primary and secondary bile acids [27].

### 2.3. Farnesoid-X-Receptor (FXR)

Farnesoid X receptor (FXR), encoded by the gene NR1H4, is expressed in multiple organs, including the liver, brain, kidneys, pancreas, cardiovascular system, and throughout the gastrointestinal tract, with a major expression in the ileum. When ligated by BA, FXR fulfills its functions through its FXRα2 isoform via specific binding to ER-2 DNA motifs [28].

The most known and classical effects of FXR are involved in bile acid, fat, sugar, and amino acid metabolism. In the liver, FXR obstructs drug-induced liver injury and the onset of HCC. In other organs, it hampers tumorigenesis and favors tissues’ wellbeing [29].

In the gut, FXR plays a double role. In fact, it helps both in orchestrating the enterohepatic circulation of BA and modulating inflammation.

FXR’s abundance is increased by luminal bile acids and decreased by inflammation [30]. FXR exerts an essential role in both the absorption and synthesis of BA. FXR activation reduces bile acid uptake by downregulating the expression/activity of the apical sodium-dependent bile acid transporter (ASBT) in the enterocytes. Furthermore, once inside the enterocytes, FXR promotes the transport of BA across the cells and their excretion into the portal circulation from the basolateral side of the enterocyte by increasing both the intracellular intestinal BA-binding protein (IBABP) and the basolateral organic solute transporters alpha and beta (OST alpha and OST beta) [31,32].

FXR is also historically known to be a transcriptional repressor of bile acid synthesis. In fact, when ligated by BA, FXR through the activation of MAP bZIP transcription factor G inhibits the transcriptions of genes involved in the alternative pathway of BA syntheses, such as CYP27A1, the rate-limiting enzyme in intestinal BA synthesis, and CYP7B1. No action is exerted by FXR on CYP7A1 [33].

Furthermore, the uptake of BA in the intestine leads to the activation of intestinal FXR, which in turn regulates the synthesis and excretion of fibroblasts growth factor (FGF)19. FGF19, secreted in the portal circulation by IEC, reaches the liver, where it binds with the hepatic FGF receptor 4 (FGFR4) in a complex with beta-Klotho. The complex FGFR4-beta-Klotho suppresses the gene-encoding CYP7A1, leading to the suppression of BA synthesis [34].

Nowadays, the role of FXR in modulating inflammation at various levels and in multiple organs together with its capability of interacting with the immune system are in the spotlight.

The first experiments that showed the role of FXR in inflammatory conditions showed a negative reciprocal regulation between NF-kB and FXR. Indeed, FXR activation by FXR agonists determined the inhibition of the NF-kB inflammatory pathway in a TNFa-dependent way, whereas the hepatic activation of NF-kB suppressed FXR-mediated gene expression [26].

Emerging evidence shows that the alteration of BA metabolism detected in IBD patients may contribute to pro-inflammatory intestinal mucosa responses through its effects on both epithelial intestinal cells and immune cells. Primary and secondary BA including oxo and iso BA derivatives act on different receptors expressed not only in IECs, but also in type 3 innate lymphoid cells (ILCs), Th17 cells, macrophages/monocytes, natural killer cells (NKCs), and dendritic cells (DCs) [35]. The role of FXR in the regulation of the intestinal immune system has been proposed for the first time by P. Vavassori and A. Mencarelli, who demonstrated that FXR -/- mice were characterized by a state of intestinal inflammation with a mild/moderate cellular infiltration of the colonic mucosa, an increased number of laminae propria CD11b+ cells, and an enhanced production of cytokines compared to WT models. They demonstrated that FXR gene ablation worsens the severity of intestinal inflammation in murine models of colitis induced by the intra-rectal administration of trinitrobenzenesulfonic acid (TNBS) or by the oral administration of dextran sodium sulfate (DSS). Moreover, treating mice with the potent semi-synthetic FXR ligand 6-Ethyl CDCA attenuated the severity of colitis and immune cell activation and the expression of various pro-inflammatory cytokines, such as TNF alpha, IL-1beta, and IL-6 in WT but not in FXR -/- mice [36].

This effect was also observed in the isolated lamina propria mononuclear cells (LPMCs) obtained from patients with IBD. The presence of FXR agonists during the in vitro stimulation of LPMCs with LPS determined a reduced production of pro-inflammatory cytokines, such as INFy, IL17, and TNFa, compared to the absence of FXR agonists [37]. Furthermore, in mice with DSS-induced colitis, FXR activation decreases local IL1β and increases systemic IL10 expression [38].

Besides the direct regulation of the immune response, FXR also interacts with toll-like receptors (TLRs). Particularly, it has been demonstrated that the activation of membrane TLRs (ex TLR2, 4, 5, and 6) reduced, whereas the activation of intracellular TLRs (ex TLR3, 7, 8, and 9) upregulated the expression of FXR via Interferon regulated Factor 7 (IRF7) [39].

Duboc et al. demonstrated that SBA but not PBA inhibited IL1b-induced IL8 secretion in Caco-2 human colon adenocarcinoma cell lines [24].

Haiping Hao et al. discovered the negative regulation mediated by FXR to the NLRP3-dependent inflammasome activation via physical interaction with NLRP3 and caspase 1; indeed, they demonstrated that mice with an overexpression of FXR were more resistant to endotoxemic shock. NLRP3 inflammasome complex has recently been implicated in IBD pathogenesis with a suspected pro-inflammatory effect; many susceptibility polymorphisms have been associated with the NLRP3 inflammasome IL-1b/18 axis, including NF-kB, CARD8, IL18, IL1b, and the NLRP3 gene itself [40].

Other studies revealed the capability of bile acids of inhibiting the innate immune system by the inhibition of NLRP3-dependent inflammasome activation [41].

Bile acids and FXR also play an important role in maintaining intestinal barrier integrity. Previous studies have demonstrated that the obstruction of bile flow through bile duct ligation leads to bacterial overgrowth and mucosal injury with increased intestinal permeability and high-rate bacterial translocation to mesenteric lymph nodes [37]. These effects could be mitigated by the oral administration of bile acids. Moreover, Inagaki et al. noted that FXR knock-out mice models have reduced epithelial barrier integrity compared to wild-type mice and a higher incidence of bacterial overgrowth. These results suggest a primary involvement of FXR in the modulation of intestinal permeability mediated by BA. The importance of FXR signaling is enhanced by the finding that in WT mouse models with bile duct ligation, the administration of GW4064, an FXR ligand, determined the improvement of intestinal barrier integrity and reduced bacterial translocation [42].

Indeed, recently it has been seen that FXR mediates an increased expression of ZO-1, Occludin, and claudin-1, proteins involved in the formation of tight junctions [43]. Furthermore, FXR induces the expression of defensins (beta-defensin-1, in particular), a molecule involved in reducing the intestinal bacterial load [44].

### 2.4. Takeda G-Protein-Coupled Receptor 5 (TGR5)

GPBAR1 (or TGR5) belongs to the superfamily of G-protein-coupled receptors (GPCR); its expression is higher in the distal ileum and colon and it is not expressed in the liver. Its natural ligands are LCA > DCA > CDCA > UDCA > CA. In the past few years, many studies have tried to investigate the role of this receptor in modulating intestinal inflammation.

Particularly, an anti-inflammatory effect of TGR5 activation by ligands such as BAR 501 through a reduction in IL6, TNFa, and INFy, an increase in IL10, and the promotion of macrophage differentiation from the M1 to the M2 phenotype has been observed in mice models of colitis. WT mice were treated with BAR 501 rescued from intestinal inflammation in a TGR5-dependent way. TGR5 activation, in fact, could promote the shift of M1 pro-inflammatory colonic macrophages towards an anti-inflammatory M2 phenotype. TGR5 activation determines the activation of protein kinase A (PKA), which then phosphorylates the transcription factor, cAMP response element-binding protein (CREB); phosphorylated CREB inhibits the transcription of the pro-inflammatory cytokines (TNFa and IL1b) while inducing those of the anti-inflammatory cytokines such as IL-10. This effect is responsible for the amelioration of the inflammation observed in TNBS colitis mouse models [35].

Similar findings have been recently noted by Sinha et al. They showed that rectal SBA administration ameliorated inflammation in DSS, TNBS, and CD45RB (hi)T cell transfer colitis mouse models. They also observed a decrease in several pro-inflammatory cytokines (CXCL10, IL17A, TNFa, CCL5) in distal colonic tissues after LCA exposure in DSS colitis-inflamed mice [45]. This anti-inflammatory effect of LCA seemed to depend on TGR5 receptor expression in immune cells rather than in epithelial cells, and especially in macrophages, as different works suggested. In particular, Yoneno et al. found that TGR5 is highly expressed in a subset of macrophages, which closely resemble the pro-inflammatory intestinal CD14+ macrophage subtype implicated in IBD pathogenesis. In these immune cells, the activation of TGR5 mediated by LCA and DCA inhibits TNFa production following the exposure to pro-inflammatory bacterial antigens through the cAMP-NF-kB pathway. In the same study, it was speculated that TGR5 agonists, such as deoxycholic and lithocholic acid, could inhibit TNFa production after bacterial LPS stimulation with a cAMP-NF-kB-dependent pathway. LPMCs isolated from the inflamed mucosa of Crohn’s Disease patients had an increased expression of TGR5 compared to LPMCs from non-inflamed mucosa or healthy controls, and the use of TGR5 agonists could inhibit TNFa production [46].

TGR5 also seems to play an important role in maintaining intestinal barrier integrity. Murine models deficient in TGR5 developed abnormal morphology of the colonic mucosa and increased intestinal permeability comprising an altered molecular architecture of epithelial thigh junctions with increased expression and abnormal distribution of zonulin 1 [47].

TGR5 is also involved in intestinal motility. It is expressed on enteric neurons and mediates the effects of BA on colonic motility, as demonstrated by the severe increase in intestinal transit in GPBAR−/− mice in comparison with controls [48].

Due to these immunoregulatory effects, TGR5 could exert an important role in the pathogenesis of IBD. Interestingly, mutational analysis of TGR5 has revealed a robust association between a single nucleotide polymorphism (rs11554825) and the US and primary sclerosing cholangitis [49].

The main effects elicited by FXR and TGR5 are summed up in Figure 2.

### 2.5. Other BARs

Retinoid-related orphan receptor-γt (RORyt) is expressed by Th17 lymphocytes and the innate lymphoid cell group 3 (ILC3). Although the ROR-yt-dependent activation of ILC3 exerts a protective action on the intestinal mucosa, Th17 activation and IL17 production have a pro-inflammatory effect [50].

A recent work by Hang et al. discovered two distinct derivatives of LCA: 3-oxoLCA and iso-alloLCA. These molecules act as T-cell regulators. In fact, through different pathways, they both present an anti-inflammatory action. 3-OxoLCA inhibits the differentiation of TH17 cells by directly binding to RORγt; isoalloLCA increases the differentiation of Treg cells through the production of mitochondrial reactive oxygen species, which leads to increased expression of FOXP3 [51].

Both these metabolites were absent in GF mouse models suggesting their derivation from microbiota. Furthermore, in mouse models of colitis, this binding decreases IL17 production and Th17 cell numbers, leading to a reduction in intestinal inflammation [52].

Paik et al. identified 12 human gut bacterial genera and their corresponding enzymes required to convert LCA to 3-oxoLCA and isoLCA: *Adlercreutzia*, *Bifidobacterium*, *Enterocloster*, *Clostridium*, *Collinsella*, *Eggerthella*, *Gordonibacter*, *Monoglobus*, *Peptoniphilus*, *Phocea*, *Raoultibacter*, and *Mediterraneibacter* [53].

PXR is a nuclear receptor that, along with many endogenous and xeno-compounds, also binds LCA. Its activation determines an anti-inflammatory effect in an NF-kB-dependent manner. PXR is expressed in human CD4+, CD8+ T lymphocytes, CD19+ B lymphocytes, and CD14+ monocytes; it is activated also by the antibiotic rifaximin. Its activation inhibits the proliferation of T lymphocytes and the expression of CD25 and INFy in vitro but also decreases the expression of IL6, TNFa, and IL8 mRNAs while promoting the expression of TGFb in an NF-kB dependent manner [54,55].

Sphingosine 1-phosphate receptor 2 (S1PR2) is expressed both in the ileum and in the colon. It is activated by conjugated primary bile acids (glycolic acid (GCA), taurocholate (TCA), glycochenodeoxycholic acid (GCDCA), taurochenodeoxycholate (TCDCA)). Recent data suggest that its inhibition exacerbates intestinal inflammation through the increased proliferation of CD4+ T lymphocytes via the extracellular-signal-regulated-kinase (ERK) pathway and decreased Zonulin-1 expression, leading to increased intestinal permeability [56].

Vitamin D Receptor (VDR) is a nuclear receptor involved in the regulation of human metabolism, immunity, and cancer. It is activated mainly by 1.25-dihydroxyvitamin D but also by LCA and its metabolites: 3-oxoLCA, 3-ketoLCA, LCA acetate, LCA propionate, and iso-alloLCA. Unlike FXR, VDR is not activated by PBA such as CDCA and CA. VDR plays an important role in the modulation of innate and adaptative immune systems [27]. It is expressed in bone, skin, kidney, intestine, and leucocytes [57]. In the gut, VDR activation by LCA induces the expression of CYP3A that metabolizes toxic LCA, probably to protect the liver from the overflow of LCA into the enterohepatic circulation [27]. VDR activation reduces the ongoing proliferation of B and T lymphocytes [58], induces activated B cell apoptosis [59], inhibits Ig production by normal human peripheral blood mononuclear (PBM) cells [60], promotes a shift from the Th1 to the Th2 phenotype through increased production of the transcription factors c-maf and GATA-3, an inhibition of INF-gamma production, and an increase in IL4, IL5 and IL10 production [61]. Previous studies have demonstrated that the oral administration of 1.25-dihydroxyvitamin D(3) enhances the suppressive capacity of CD4+, CD25+, and Foxp3+ T-regulatory lymphocytes [62], renders dendritic cells tolerogenic, promotes the differentiation of regulatory T cells [63], and inhibits IL17 production and Th17 cell differentiation through RORyt [64].

In murine models, calcitriol administration determinates the reduction of different inflammatory cytokines, such as IL1, IL6, IL8, IL12, IL23, IL17, and TNFa, while inducing TGFb, FOXP3, and IL10 production [65]. In addition, murine models with the transgenic expression of human VDR were protected from DSS colitis and had a reduced IEC apoptosis; therefore, VDR may also be involved in the regulation of IEC homeostasis and the maintenance of intestinal mucosal barrier integrity [66].

Several studies have associated serum low vitamin D levels with poor prognosis in IBD, such as an increased risk of surgery and clinical relapse within 12 months, independent of the endoscopic or histologic grade [67,68]. In addition, in a meta-analysis by Le-Ning Xue et al., some VDR polymorphisms have been associated with an increased (the tt genotype of TaqI, ff genotype of FokI, and the tt genotype of ApaI) or a reduced (“a” allele carrier status of ApaI) susceptibility to the development of CD or UC [69].

## 3. Bile Acid-Induced Diarrhea (BAD)

### 3.1. Classification

BA, CDCA, and DOA, in particular, are able to stimulate intestinal motility when they come into contact with intestinal mucosa, especially if they are rich in high concentrations (3 to 5 mmol/L), thus causing the so-called BA-induced diarrhea [70]. The onset of diarrhea, especially in patients with underlying IBD and ileo-cecal resection, is a confounding factor that often complicates the clinical approach and management.

Bile acid-induced diarrhea has historically been divided into four subtypes [71].

Type 1 is due to ileal dysfunction or resection, as we often observe in Crohn’s Disease. The ileum is the site where most of the BA are reabsorbed. Ileal dysfunction or resection limits the enterohepatic circulation, thus allowing a greater amount of BA to reach the colon and generate diarrhea [72]. This subtype, as mentioned above, is the one most frequently involved in IBD.

In a rat model of colitis, the authors revealed that during the acute phase of colitis, the rats had a reduced expression of apical sodium-dependent bile acid transporter (ASBT) in the ileum, with a concomitant increased excretion of BA in feces as well as diarrhea [73]. These results suggest the involvement of ASBT expression in maintaining BA homeostasis. The increased number of BA in the colon due to their biochemical structure stimulates increased secretion of electrolytes/water and increased mucosal permeability, as well as causes high-amplitude propagating contractions arising in the proximal colon leading to the development of diarrhea [74].

Van den Bossche et al. in 2017 investigated the expression of ASBT in the TNF^ΔARE/WT^ murine model of ileitis, where the intestinal inflammation depends on the pro-inflammatory cytokine TNF-alfa. This model, according to the authors, was a more appropriate IBD model for studying BA metabolism alterations since >95% of the bile acid pool is efficiently reabsorbed in the distal ileum, therefore, it does not reach the colon. Using this model, the authors showed a reduced expression of ASBT and Ostα and Ostβ in mice with ileitis. These results were confirmed by the investigators of Caco-2 cell monolayers exposed to TNF-alfa. Moreover, the oral administration of tauroursodeoxycholic acid (TUDCA), a secondary bile acid, to mice attenuated the clinical and histopathological parameters of murine ileitis and alleviated the downregulation of bile acid transporters in these mice [75].

Decreased ASBT expression has also been described in canine, rabbit, and mouse models of gut inflammation. Dogs with chronic inflammatory enteropathy (CIE) have decreased ileal ASBT protein expression. Giarretta et al. found a significant negative correlation between ileal ASBT expression and the cumulative histopathological score of ileal damage. Additionally, dogs with CIE had an increased fecal dysbiosis index, with an increased percentage of primary BA in feces compared to controls [76]. Similar results have been detected by Wang et al. in dogs with chronic enteropathy. They showed a reduction in SBAc (LCA and DCA) in feces associated with intestinal dysbiosis. Treatment with hydrolyzed protein reduced the amount of pathogenic bacterial species (*E. coli*, *C. perfrigens*) while increasing the levels of BA-producing bacteria (Clostridium hiranonis) in the feces. These results were linked to the clinical remission of sick dogs [77].

BAD type 2 is an idiopathic form. It is thought to be related to an altered functioning of FGF19. This transcription factor, which acts as a repressor of bile synthesis and is normally secreted by ileal epithelial cells, appears to be reduced in patients with this kind of condition. This situation leads to an overexpansion of the BA pools that enter the colon [78].

Type 3 includes a variety of conditions, including chronic pancreatitis, microscopic colitis, cholecystectomy, SIBO, and radiation enteritis. A full understanding of the pathogenetic mechanisms is yet to be achieved. The most common among this group of conditions is post-laparoscopic cholecystectomy diarrhea [79]. In this condition, with the lack of a reservoir for bile and BA, an increased amount of BA comes into contact with intestinal mucosa, thus causing diarrhea [80].

The fourth type of BA-induced diarrhea is a rare condition usually diagnosed in the pediatric age group. It is characterized by a deficit in BA reabsorption at the ileal level due to a congenital deficit of the sodium-dependent bile acid transporter gene (SLC10A2) [81].

A recently published study from Camilleri et al. compared the molecular and biochemical parameters in 161 patients with irritable bowel syndrome and 44 patients with biliary acid diarrhea. Patients with BAD showed lower alpha and beta diversity with an increased amount of Firmicutes and decreased expression of bile acid thiol ligase (involved in the transformation of primary to secondary BA) as well as decreased sulfatases. Furthermore, in the same study, biopsies from patients with BAD presented downregulation of BA transporters (SLC44A5 in particular) and upregulation in barrier-weakening genes (such as CLDN2), together with increased inflammatory activity within the lamina propria [82].

### 3.2. Diagnosis

The insurgence of BA-induced diarrhea is a common occurrence in inflammatory bowel disease, especially in Crohn’s disease. In fact, especially after ileo-caecal or ileocolonic resections, bile acid diarrhea represents the most common cause of non-inflammatory diarrhea [83].

The onset of diarrhea after ileo-caecal resection in such patients represents a significant diagnostic and therapeutic challenge for the clinician. In fact, diarrhea could be provoked by disease reactivation, small bowel bacterial overgrowth [84], irritable bowel syndrome [85], or BA malabsorption itself.

A useful, low-cost, and accessible first-level diagnostic tool in these cases is represented by the measurement of fecal calprotectin. High levels (>100 micrograms/g of stool) of fecal calprotectin correlate with the presence of intestinal inflammation and therefore suggest a disease reactivation with sufficient sensitivity and specificity [86,87]. Low levels of fecal calprotectin, on the other hand, prompt the physician to investigate other causes of diarrhea.

When suspecting BA-induced diarrhea, a trial with cholestyramine is usually successful in clarifying the origins of the disturbance. Though, over the years, novel as well as old diagnostic tools have been perfected.

The first tool is the dosage of the blood concentration of C4 (7α-hydroxy-4-cholesten-3-one). C4 is an intermediate in bile acid synthesis; therefore, when BA are being lost in the colon and their hepatic synthesis increases, serum levels of this intermediate consequently increase [88].

A second, potential, diagnostic tool is represented by the dosage of serum levels of FGF19. As mentioned above, FGF19 is implicated in regulating the synthesis of BA. In patients with BA-induced diarrhea, FGF19 appears to be a good predictor of BAD occurrence [89].

A recent publication by Battat et al. investigated the relationship between serum C4 and BAD in patients with Crohn’s disease. They observed significantly increased serum concentrations of C4 with low serum levels of FGF19, an inhibitor of BA synthesis, in CD patients who underwent an ileal resection, compared to patients with ulcerative colitis. The concentration of C4 correlated with daily liquid bowel movements and inversely correlated with FGF19. These findings suggest the huge potential of C4 as a biomarker to be used in clinical practice to discriminate between CD patients with BAR-dependent diarrhea and those with active CD [90].

Besides inflammatory bowel disease, BAM is common also in microscopic colitis, both in lymphocytic colitis and collagenous colitis, as different studies have demonstrated over the years [91]. In a recent study, the authors investigated the diagnostic accuracy of C4 and compared it with FGF19 as a biomarker for the diagnosis of BAD. They enrolled chronic diarrhea patients with active IBD, inactive IBD, IBD after surgery, IBS, microscopic colitis, and healthy subjects. Serum levels of 7α-hydroxy-4-cholesten-3-one > 48.9 ng/mL and of FGF19 < 60 pg/mL detected bile acid malabsorption with 82.6% sensitivity and 84.3% specificity in patients with chronic diarrhea. The median level of C4 in active IBD patients was 53.1 ng/mL, IBD remission was 52.2 ng/mL, IBD after surgery was 85.7 ng/mL, IBS-D was 7.5 ng/mL, microscopic colitis was 69.3 ng/mL, and healthy controls was 3.7 ng/mL. The authors demonstrated that both C4 and FGF19 can be used with high diagnostic accuracy as screening biomarkers for bile acid malabsorption in microscopic colitis and inflammatory bowel disease [92].

### 3.3. The seHCAT Test

The third diagnostic tool, which came into the spotlight during the 80s, especially in Anglo-Saxon countries, is the SeHCAT test. This exam uses the selenium-labeled bile acid SeHCAT to assess the integrity of enterohepatic circulation. It aims to measure and quantify the ileum’s capacity to reabsorb BA. This test must be executed with a previous 7-day suspension of cholestyramine [93].

One of the pioneer studies on SeHCAT was performed in 1994 by Nyhlin et al. and 53 patients with Crohn’s disease were enrolled. Among the patients who had previously undergone an ileo-cecal resection, 93% had a positive test for BA malabsorption; on the other hand, among the patients without an ileo-cecal resection, only 28% had a positive test for BA malabsorption [94].

The SeHCAT test has also been used to detect the association that exists between collagenous colitis and bile acid-induced diarrhea. One study found that 44% (12/27) of patients with collagenous colitis had bile acid-induced diarrhea when investigated through SeHCAT testing. Long-term follow-up (median 4.2 years) revealed a rapid improvement in diarrhea after treatment with cholestyramine in 21/27 patients: 11/12 of patients with BAM and 10/15 of patients with a normal (75) SeHCAT test [95]. Although the association between bile acid diarrhea and lymphocytic colitis is less clear, these patients have lower 75SeHCAT values (range 1.7–53, median 24%) than the control subjects (range 8–91, median 38%) [96]. These results suggest that disturbed bile acid homeostasis is a feature of microscopic colitis.

SeHCAT is thought to play a role in predicting the response to treatments with bile acid. In a retrospective analysis of patients tested with SeHCAT between 2008 and 2014, a positive 75SeHCAT predicted a good or partial response to BAS of 66.7% (mild), 78.6% (moderate), or 75.9% (severe BAD). Further to CD, the factors strongly associated with the occurrence of BAD were cholecystectomy and ileal resection from other causes [97].

In more recent times, a systematic metanalysis investigated the cost effectiveness of performing SeHCAT in two different populations (patients with IBS-like diarrhea and patients with Crohn’s disease) without achieving significant conclusions [98].

### 3.4. Treatment and Therapeutic Perspectives

Symptomatic management with cholestyramine has been a milestone in the therapeutic approach to BA-induced diarrhea and still represents the first therapeutic choice for patients with suspected or proven BAD. Cholestyramine is undoubtedly effective in reducing the symptoms and clinical manifestations of BAD [99]. However, cholestyramine formulations could often result in unpalatable and other side effects (constipation, nausea, borborygmi, flatulence, bloating, and abdominal pain), thus compromising patients’ adherence to therapy [100]. For this reason, new formulations and new molecules have been developed such as colesevelam or colestipol [101].

Even though cholestyramine plays a central role in the management of BAD in IBD and other similar conditions, new therapeutic horizons are coming into the spotlight. Among these, the most interesting and studied agents are the FXR agonists. In fact, FXR has properties that make it an attractive target for treating BAD. First, it inhibits epithelial colonic fluid secretion, and second, it downregulates hepatic bile acid synthesis through FGF19 [102].

The most promising agent belonging to the category of FXR agonists is, without doubt, obeticholic acid (OCA). This agent was first approved in 2016 for the treatment of UDCA-refractory CBP [103] and then tested with good results for the treatment of nonalcoholic steatohepatitis [104].

Lately, increasing attention is being focused on the use of OCA for the treatment of BAD. Moreover, it is interesting to note that nonalcoholic fatty liver disease (NAFLD) is somewhat associated with an imbalance in the BA cycle, dysregulation in the FXR-FGF19 axis, and consequent diarrhea, as demonstrated in an interesting study by Appleby et al. [105].

An increasing amount of data is being produced in the literature on the use of OCA in treating bile-induced diarrhea, both in experimental models and clinical practice.

Both in CaCo cells and mice, FXR agonists have shown their potential activity as antisecretory agents [106]. This feature had already been investigated for UDCA with similar results [107].

In 2018 Hvas described the case of a 32-year-old woman with quiescent Crohn’s disease presenting diarrhea and diagnosed with severe bile malabsorption. In the light of refractoriness to conventional therapies with loperamide, octreotide, and PPIs, a successful attempt with OCA was performed with an immediate improvement in diarrhea. Curiously, after the cessation of OCA, diarrhea immediately started again [108].

Walters et al. performed a small clinical trial on 28 patients divided into three groups (primary BAD, secondary BAD, and idiopathic BAD). The administration of OCA 25 mg per day for 2 weeks led to a significant increase in stool frequency and form, especially in patients with primary BAD and with secondary BAD and shorter ileal resections. The improvement in diarrhea was witnessed by an increase in serum FGF19 and a reduced serum C4 [109].

Notwithstanding interesting perspectives on the management of BAD, especially in patients with IBD and previous ileal resections, OCA is not free of side effects and contraindications. In particular, it has to be remembered that OCA administration must be limited or avoided in patients with severe hepatic dysfunction (CPT B or C cirrhosis). Furthermore, the administration of OCA is associated with annoying side effects such as pruritus [103].

For this reason, in recent times, new FXR agonists are coming into the spotlight and are being tested to find an alternative to OCA. Among these, an interesting perspective is offered by tropifexor [110].

The first studies in humans led to interesting results with no safety issues with the administration of up to 3000 µg daily of tropifexor [111]. In a double-blind, multicenter, randomized, cross-over study, patients received 60 µg of tropifexor. The molecule showed good data in terms of safety and tolerability, with no patients complaining of pruritus during the period of drug administration [112].

Last but not least, a significant role in modulating the metabolism of BA could be played by probiotics. In fact, several studies have shown the beneficial properties of different probiotics. These effects are mostly related to the deconjugation of BA operated by bile salts hydrolase (BSH). Interestingly, the deconjugation of BA is responsible for the hypocholesterolemic effect of probiotics [113]. Particularly, two probiotics currently used in clinical practice must be mentioned. The probiotic mixture VSL#3 promotes BA deconjugation and fecal excretion, with subsequent downregulation of the gut–liver FXR-FGF15 axis and therefore increased neo-synthesis of BA within the liver [114]. Aside from the increased consumption of cholesterol for BA neo-synthesis, we cannot exclude the role played by free amino acids (taurine or glycine) liberated from BA deconjugation. Indeed, glycine acts through the glycine receptor chloride channel to inhibit neurotransmission, whereas taurine has effects on the GABA_A_, GABA_B_, and glycine receptors. Moreover, taurine performs several beneficial functions both as an antioxidant and anti-inflammatory agent [115].

Combinations of *Lactobacilli* have shown to be effective in improving IBS symptoms [116].

*Clostridium scindens* is notoriously associated with protection against *C. difficile* infection. Charlie G. Buffie et al. hypothesized that this protective mechanism may derive from the expression of crucial enzymes for secondary BA synthesis, represented by bai operon genes, such as bile acid 7alpha-hydroxysteroid dehydrogenase enzyme, which are possessed by an extremely small fraction of intestinal bacteria. Most bile acid 7-dehydroxilating-bacteria are cluster XIVa *Clostridia* and this cluster could represent a probiotic candidate that corrects a clinically relevant dysbiosis [117].

### 3.5. The Role of Faecal Microbiota Transplantation (FMT) in Bile Acid Metabolism

Finally, fecal microbiota transplantation (FMT) also deserves our attention in the regulation of BA metabolism and consequent BAD. A study by Seekatz [118] et al. investigated the changes in bile acids within FMT-treated patients. After FMT, they observed a significant shift in the bile acid profile, with a decrease in the taurine-conjugated bile salts, TCA and TCDCA, the unconjugated bile acid chelates and some glycine-conjugated bile salts, and an increase in the secondary bile acids DCA, LCA, and UDCA. Particularly, the levels of secondary bile acids surpassed the concentrations observed in donors, with a drastic variation over the recovery period from one time point to another suggesting the dynamic recovery of microbial-derived secondary BAs. These results had already been described in patients treated with FMT for clostridium difficile infection (CDI) in a previous study. The authors observed higher concentrations of secondary bile acids in post-FMT fecal samples in contrast with the predominance of primary bile acids and bile salts in pre-FMT fecal samples. The authors concluded that the metabolism of bile acids is disrupted in patients with recurrent CDI, probably due to the important dysbiosis of these patients, which makes the microbiota incapable of metabolizing primary bile acids; FMT corrected this abnormality. The normalization of BA metabolism could contribute to the resolution of mild diarrhea symptoms experienced by patients treated with standard therapy for CDI prior to FMT. On the other hand, lithocholic acid and other secondary bile acids, which were almost undetectable in pre-FMT fecal samples, inhibit C. Difficile germinations and colony growth, thus contributing to the resolution of the infection [119].

## 4. Conclusions

Bile acid metabolism and the interaction between bile acids and bile acid receptors both play a central role both in worsening diarrhea symptoms and in determining the pathogenesis of inflammatory bowel diseases including microscopic colitis.

The altered synthesis and metabolism of BA, in fact, exerted a pro-inflammatory effect on intestinal mucosa through the FXR and TGR5 receptors via several mechanisms, thus determining the increased inflammatory response.

On the other hand, especially in patients with definite predisposing anatomical factors, BA are responsible for the occurrence of bile acid-induced diarrhea, a clinical entity often underestimated and undertreated.

The combination of these factors gives new perspectives and new insights into the role of bile acids in the pathophysiology of IBD and microscopic colitis. A better comprehension of the mechanisms related to BA inflammation and diarrhea could give new therapeutic and diagnostic solutions for improving the clinical outcomes and patients’ quality of life.

## Figures and Tables

**Figure 1 nutrients-14-02664-f001:**
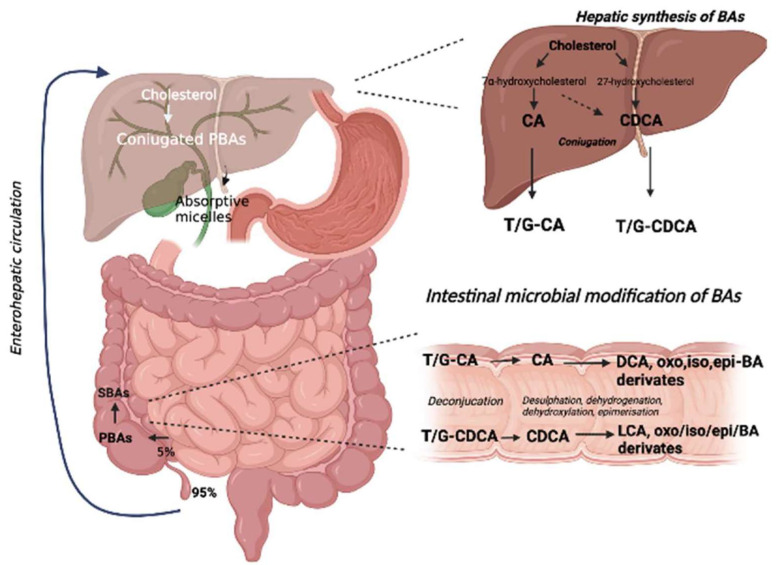
Biotransformation of biliary acids. BA metabolism starts with cholesterol and is converted into PBA in the liver (CA and CDCA). Subsequently, these are conjugated with glycine or taurine (T/G-CA and T/G-CDCA), stored in the gallbladder, and released in the intestine. Here, the conjugated BA are first deconjugated by intestinal bacteria and then metabolized through a series of chemical reactions (desulfation, dehydrogenation, dihydroxylation, and epimerization) catalyzed by bacterial enzymes to other types of SBA (DCA and LCA and oxo, iso, epi-BA derivates).

**Figure 2 nutrients-14-02664-f002:**
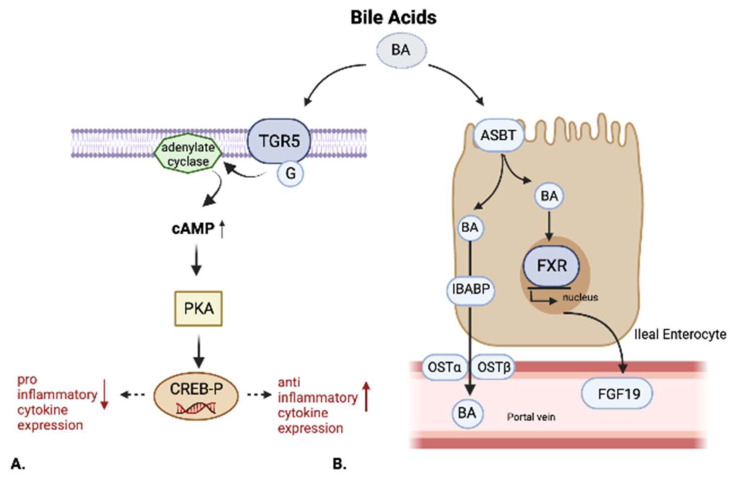
The effects of BA through TGR5 and FXR receptors. (**A**) BA actives TGR5(Takeda G-protein-coupled receptor 5) and determines the activation of PKA (protein kinase A), which phosphorylates CREB (cAMP response element-binding protein). CREB inhibits the transcription of pro-inflammatory cytokines and induces those of anti-inflammatory cytokines. (**B**) The ASBPT (sodium-dependent bile acid transporter) allows the entrance of BA (bile acids) into the ileal enterocyte followed by the activation of FXR (farnesoid-X- receptor), which regulates the synthesis and expression of FGF19 (fibroblasts growth factor). It suppresses BA synthesis. The transport and the excretion of BA in the portal circulation are done by IBABP (intestinal bile acid-binding protein) and OSTα and OSTβ (organic solute transporter).

## Data Availability

Data available in a publicly accessible repository. The data presented in this study are openly available in Pubmed.

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
