# Peer review of "Bile Acid-Related Regulation of Mucosal Inflammation and Intestinal Motility: From Pathogenesis to Therapeutic Application in IBD and Microscopic Colitis"

_nutrients, 2022, doi:10.3390/nu14132664_

Round 1
Reviewer 1 Report
I've read with attention the paper of Federica Di Vincenzo et al. that widely described the relation between bile acid metabolism, IBD, and the gut microbiome.
The aim of the review has been clearly defined. Overall, the manuscript is logically organized and good written in English, but in the manuscript I detected lots of typos /listed below/.
The work is well organized and comprehensively described.
Below is my comment to improve the manuscript:
Page 2, line 44 - why the phrase "bile acids" are written with an underline?
Page 2, lines 44-51 - Please check the format of the text
Page 2 , line 54 and 60 and 69, and 74, and 82- missing spaces between text and []
Page 2, line 55 -> It is not entirely clear what "their" refers to.
Page 2 -> Missing [10] between [9] and [11].
Page 2, line 67 -> Please to change [6], [7] to [6,7]. The same situation e.g. in page 11 line 466
Page 3, Figure 1. -> I'm not sure if the list of abbreviations is presented in the form acceptable to the journal.
Page 3, line 101, 102, 104-> Please use the abbreviation BA consequently
Page 4, line 125, 128, 132, 134, 154, 158 -> missing space between text and []. Please check this error throughout the manuscript.
Page 4, line 136 -> Please change " shorty fatty acids" to "short-chain fatty acids"
Page 4, line 140 -> the abbreviation "CD" was not explained before
Overall, the work is comprehensively described but I have slight lack of information about probiotics supplementation (in the section "treatment and therapeutic perspective). Please consider adding a short fragment about the possibilities of probiotic therapy.
Author Response
Comments and Suggestions for Authors:
I've read with attention the paper of Federica Di Vincenzo et al. that widely described the relation between bile acid metabolism, IBD, and the gut microbiome.
The aim of the review has been clearly defined. Overall, the manuscript is logically organized and good written in English, but in the manuscript I detected lots of typos /listed below/.
The work is well organized and comprehensively described.
Below is my comment to improve the manuscript:
- Page 2, line 44 - why the phrase "bile acids" are written with an underline?
Response: Thank You for pointing out this inaccuracy. We corrected the error in the manuscript.
- Page 2, lines 44-51 - Please check the format of the text
Response: Thank You for this suggestion. We checked the format of the text in lines 44-51.
- Page 2 , line 54 and 60 and 69, and 74, and 82- missing spaces between text and []
Response: Thank You for pointing out this inaccuracy, we added the missing space throughout the whole text.
- Page 2, line 55 -> It is not entirely clear what "their" refers to.
Response: In line 55 “their” is referred to bile acids. We edited the text to make it clearer.
- Page 2 -> Missing [10] between [9] and [11].
Response: Thank You for pointing out this inaccuracy, we corrected the order of all references.
- Page 2, line 67 -> Please to change [6], [7] to [6,7]. The same situation e.g. in page 11 line 466
Response: Thank You for pointing out this inaccuracy, we corrected the way we referenced throughout the whole text.
- Page 3, Figure 1. -> I'm not sure if the list of abbreviations is presented in the form acceptable to the journal.
Response: We thank the reviewer for pointing out this inaccuracy. We changed the presentation of abbreviations in Figure 1.
- Page 3, line 101, 102, 104-> Please use the abbreviation BA consequently
Response: We thank You for this suggestion, we are now using the abbreviation BA in lines 101, 102, 104.
- Page 4, line 125, 128, 132, 134, 154, 158 -> missing space between text and []. Please check this error throughout the manuscript.
Response: Thank You for pointing out this inaccuracy, we checked this error and added the missing space throughout the whole manuscript.
- Page 4, line 136 -> Please change " shorty fatty acids" to "short-chain fatty acids"
Response: We thank the reviewer for pointing out this inaccuracy. As You suggested, we changed “shorty fatty acids” to “short-chain fatty acids”.
- Page 4, line 140 -> the abbreviation "CD" was not explained before
Response: We thank You for pointing out this inaccuracy. The abbreviation CD was referred to Crohn’s Disease. We explained it in the text (line 141).
- Overall, the work is comprehensively described but I have slight lack of information about probiotics supplementation (in the section "treatment and therapeutic perspective). Please consider adding a short fragment about the possibilities of probiotic therapy.
Response: We thank you for this precious comment which let make our manuscript more interesting and complete for the readers. We edited the paper (lines 583-605) discussing the possible role of probiotic therapy in modulating bile acid metabolism.

Reviewer 2 Report
I would not start the first paragraph with this sentence, please reconsider Bile composition is not limited only to bile acids.
Missing reference An imbalance on bile production, reabsorption or functioning is associated with various gastrointestinal affections. please add
Line 54, add space before reference, and onwards
In Figure 1, this is a weird presentation of abreviations, please reconsider
This is not correct way of referencing the content of conjugated bile acid and of 3-OH-sulfate bile acids[16], [17]., consider [16,17]
Names of pathogens should be in cursive.
Author Response
Comments and Suggestions for Authors:
- I would not start the first paragraph with this sentence, please reconsider Bile composition is not limited only to bile acids.
Response: We thank the reviewer for this generous suggestion. We corrected the sentence as You suggested.
- Missing reference An imbalance on bile production, reabsorption or functioning is associated with various gastrointestinal affections. please add
Response: We apologize with the reviewer for this lapse. We added it in the manuscript.
- Line 54, add space before reference, and onwards
Response: Thank You for pointing out this inaccuracy, we corrected the way we referenced throughout the whole text.
- In Figure 1, this is a weird presentation of abreviations, please reconsider
Response: We thank the reviewer for pointing out this inaccuracy. We changed the presentation of abbreviations in Figure 1.
- This is not correct way of referencing the content of conjugated bile acid and of 3-OH-sulfate bile acids[16], [17]., consider [16,17]
Response: Thank You for pointing out this imprecision, we corrected the way we referenced throughout the whole text.
- Names of pathogens should be in cursive.
Response: Thank You for pointing out this inaccuracy, we apologize with the reviewer. We corrected names of pathogens in cursive in the whole manuscript.
